



# 1 Measurement report: Determination of Black Carbon concentration
# 2 in PM$_{2.5}$ fraction by Multi-wavelength absorption black carbon
# 3 instrument (MABI)

Anna Ryś[1], Lucyna Samek[1]
[1]Faculty of Physics and Applied Computer Science, AGH University of Science and Technology, 30 Mickiewicza Ave,
Krakow, 30-059, Poland
*Correspondence to*: Lucyna Samek (lucyna.samek@fis.agh.edu.pl)
**Abstract.** The evaluation of black carbon (BC) sources is very important, especially in environmental sciences. This study
shows how the contributions of biomass burning and fossil fuel/traffic to PM$_{2.5}$ mass can be assessed. MABI was used for this
purpose and gave the possibility to measure the transmission of light at different wavelengths. Absorption coefficients were
calculated from measurements data and recalculated for concentrations of eBC. The samples of PM$_{2.5}$ fraction were collected
from February 1, 2020 to March 27, 2021 every third day in Krakow, Poland (50°04'N, 19°54'47" E). The concentrations of
equivalent BC (eBC) from fossil fuel/traffic and biomass burning were in the range 0.82-11.64 $\mu g\ m^{-3}$ and 0.007-0.84
$\mu g\ m^{-3}$, respectively. At the same time, PM$_{2.5}$ concentrations varied from 3.14 to 55.24 $\mu g\ m^{-3}$. It means that about 18 % of
PM$_{2.5}$ mass belongs to eBC and 11.3 % of this value comes from biomass burning. The eBC contribution is the significant part
of PM$_{2.5}$ mass and we observed seasonal variation of the eBC concentration during the year with the peak in winter. The
contribution of biomass burning to PM2.5 mass is more stable during the whole year. The eBC concentration during workdays
is a bit higher than during weekend days but biomass burning is similar for both days (work and weekend taken as the mean
for the whole period).

## 20 1 Introduction

For many years, scientists have conducted air pollution research that led to the possibility of successful identification of
numerous aerosol particular matter (PM) components harmful to the environment and as such to human health. Now, scientists
can perform a quantitative and qualitative analysis of components present in the air as organic and inorganic substances,
chemical elements, heavy metals, and black carbon.
The largest contributor to air pollution and global change is the emission of the carbonaceous particle which is produced during
the combustion of biomass and fossil fuel (Petzold et al., 2013). Black carbon (BC) is the one fraction of the carbonaceous
aerosol which is produced during incomplete combustion (Diapouli et al., 2017; European Environment Agency, 2013; Petzold
et al., 2013). BC is characterized by the strong absorbs of all wavelengths of solar radiation (Bond and Bergstrom, 2006;
Petzold et al., 2013; U.S. EPA (U.S. Environmental Protection Agency), 2012). Based on the fact that light absorption particles



can warm the atmosphere, scientists consider the BC as the second most critical reason for global warming in terms of direct
impact (Jacobson, 2001; Liu et al., 2016). The BC influences not only general climate change but also causes indirect effects
for instance it reduces local visibility and is responsible for the appearance of the brown hazes in a city (Horvath, 1993; U.S.
EPA (U.S. Environmental Protection Agency), 2012). According to the "Report to Congress on Black Carbon", most of the
global BC emissions come from Asia, Latin America, and Africa while major sources were open biomass burning, residential,
and transport (U.S. EPA (U.S. Environmental Protection Agency), 2012). Although BC has a strong harmful impact on the
environment, the lifetime of BC in the atmosphere is relativity short. The reduction of the BC can be achieved by the strategies
targeting to lower BC emissions connection with the identified sources of BC, global air pollution management, and policy-
making (Diapouli et al., 2017; U.S. EPA (U.S. Environmental Protection Agency), 2012). Since research on black carbon is
being constantly improved there is some inconsistency in terminology. Bond and Bergstrom indicated that scientists use
different names for "Black Carbon" for instance: "soot", "light absorbing carbon", "elemental carbon", "refractory carbon",
"graphitic carbon" (Bond and Bergstrom, 2006; Petzold et al., 2013). These nomenclatures are connected with composition,
optical properties, and particle morphology.
Over the years, different measurement techniques and instruments were used to describe and determine the BC mass. These
techniques are usually based on light absorption scattering or thermal radiation measurements. The instruments which are
commonly used range from the aethalometer to Multi-angle absorption photometry (MAAP) or Photoacoustic Instruments, as
well as techniques, and recommended terminology that is widely discussed in review papers by Petzold, A. et al, Bond, T. C.
& Bergstrom, R., and Lack, D. et al (Bond and Bergstrom, 2006; Lack et al., 2013; Petzold et al., 2013; Petzold and
Schönlinner, 2004).
There are a lot of publications about BC and most of them analyzed fraction $PM_{2.5}$. What is more, the Report EU presented
results of BC analysis of other countries based on fraction $PM_{2.5}$. This study was also conducted basing on the BC from fraction
$PM_{2.5}$ enabling comparison of received results with previously presented studies.
Our study aims to analyze the results of light absorption coefficients and BC mass concentrations which were obtained for
different wavelengths. The calculation is based on the Lambert-Beer law, where $\sigma_a$ is the absorption coefficients of the aerosol
particles:
$$\frac{I_0}{I} = exp(\sigma_a * X) ,$$ (1)
where $I_0$ and I are the unexposed intensity and the exposed transmission intensity, X is the length of the sampled air column
(Taha et al., 2007a).
This paper describes the methodology of calculating the results obtained by MABI step by step. Next, the results are
demonstrated like graphs and tables. The key aspect is legislative changes which are in force in Kraków which should have an
immense impact. Our measurement was conducting after that the local government implemented the ban on using solid fuel
in households stoves in September 2019 in Krakow. We are convinced, that to compare previous studies
(https://acp.copernicus.org/preprints/acp-2021-197/), our study is very valuable and it demonstrates current situation under





new circumstances. Our paper presents the most recent data, which last record was from March this year. What is more, our
manuscript especially concentrations on presenting clearly data for BC of fraction $PM_{2.5}$.
To the best of our knowledge, this is one of the first studies which includes the results of measurement by MABI for the whole
year. Therefore we were not able to compare our results presented in this paper with the results from similar studies carried
out by this instrument. However, we believe that our work will be valuable for organizations and institutes which have just
started or continue working with MABI for the comparison or verification of the results.

## 69 2 Sampling and Method

### 70 2.1 Sampling location

The sampling camping was conducted from the research station located at AGH University of Science and Technology in
Krakow (50°04'00.5" N, 19°54'46.8" E). This station is equivalent to the urban background station. Next to the sampling place,
there are housing estates and a two-lane dual carriageway. The sampling place is about 2 km from City Center.

### 74 2.2 Sampling

Sampling was performed between February 1, 2020, and March 27, 2021 for 24 hours every three days. Some of the sampling
days were omitted due to the temporary suspension of the research. $PM_{2.5}$ samples were collected on Teflon filters (GE's
Whatman, PTFE 46.2 mm, 2.0 μm), by the use of a low-volume ($2.3\ m^3\ h^{-1}$) sampler (Sequential 47/50-CD with Peltier
cooler, Sven Leckel GmbH, Berlin, Germany). The concentration of the fine fraction was determined. The mass of filters was
measured before and after sampling. The filters were stored in the following condition 20 ± 1 °C and humidity 50 ± 5 % for 24
hours before weighing. The Teflon filters were analyzed for black carbon by MABI.

### 81 2.3 Multi-wavelength absorption black carbon instrument

Multi-wavelength absorption black carbon instrument (MABI) measures light absorption at seven different wavelengths: 405
nm(UV), 465 nm, 525 nm, 639 nm, and infrared 870 nm, 940 nm, and 1050 nm. MABI was developed by the Australian
Nuclear Science and Technology Organisation (www.ansto.gov.au). This instrument consists of the optical assembly and
electronic case. The instrument optics includes, among others, the multi-wavelength light source
(7 LEDs), sampler holder, and photo-detector. What is more, it is used the opaque glass in "MABI units to scatter the scattered
light back through the filter to the detector"(David D. Cohen, 2020). The calibration, zero the detector is performed
automatically and saved by the MABI software application. All results are saved by the MABI software application as a ".csv".
Data analysis was performed using Excel.
The samples were scanned before and after sampling by MABI. MABI scans were performed on unexposed filters to determine
the absorption at each wavelength from the filter substrate and obtained data is called $I_0$. Scans were repeated on the same filter





after sampling (exposed filters) to determine the absorption at each wavelength from filter substrate and collection particles
and this data is called I. Based on these values ($I_0$ and I), the exposed filter area and sampled air volume, the black carbon light
absorption coefficients $b_{abs}$ ($Mm^{-1}$) were determined at each wavelength with the following equation:
$b_{abs} = 10^2 \cdot \frac{A}{V} \cdot ln\left[\frac{I_0}{I}\right]$,                     (2)
here $I_0$ and $I$ are the light transmission through an unexposed filter and an exposed filter. Parameter $A$ is the collection area of
the exposed filter ($cm^2$) and $V$ is the volume of sampled air on the filter ($m^3$).
The black carbon (BC) mass concentration ($ngm^{-3}$) was obtained using a mass absorption coefficient $\varepsilon$ ($m^2 g^{-1}$) at each
wavelength from:
$BC\ (ngm^{-3}) = \frac{10^5 \cdot A}{\varepsilon \cdot V} \cdot \ln\left[\frac{I_0}{I}\right] = \frac{10^3 \cdot b_{abs}}{\varepsilon}$,                     (3)
MABI Manual presents tests, results and mass absorption coefficients for different types of filters. Following the study
presented by Atanacio et al.,David D. Cohen, Taha et al (Atanacio et al., n.d.; David D. Cohen, 2020; Taha et al., 2007b), the
Equation (3) was presented below and the values mass absorption coefficient $\varepsilon$ for λ = 639 nm was assumed as $\varepsilon_{\lambda=639\,nm}$ =
6.036 $m^2\ g^{-1}$, which was recommended for 47mm Teflon filters by ANSTO. The author assumed scattering correction (C)
and loading correction (R) based on the deep discussion presented in (David D. Cohen, 2020), where C and R (C = R = 1).
The mass absorptions coefficients, for each wavelength, was estimated using the following calculation formulas, where $\lambda_2 =$
$\lambda_{639\,nm}$,:
$\varepsilon(\lambda_1) = \varepsilon(\lambda_2) \cdot gradient\left(\frac{\lambda_2}{\lambda_1}\right)$                     (4)

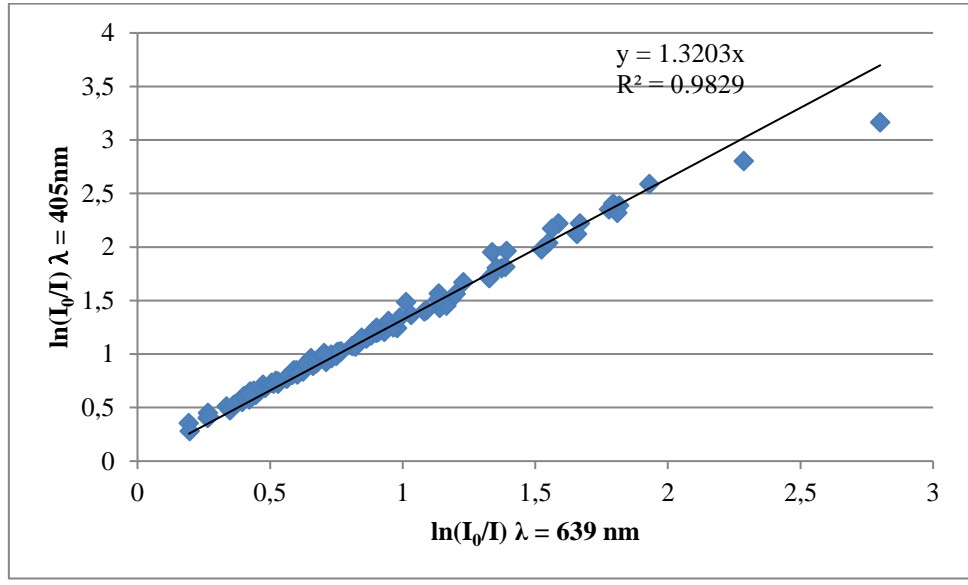


**Figure 1 The plot of ln($I_0$/I) for λ = 405nm versus ln(I₀/I) for λ = 639 nm, where $gradient\left(\frac{\lambda_2}{\lambda_1}\right) = 1.3203$, and $\varepsilon(\lambda_{405})$ =**
**7.97$m^2 g^{-1}$ for sampling from Krakow site.**



## 3 Results and discussion

The raw data obtained by MABI were calculated according to the methodology described above and were transformed to absorption coefficients (ε), equivalent black carbon (eBC) mass concentrations. The mass absorption coefficient for wavelength is presented as:

$$\varepsilon(\lambda) = a \cdot \lambda^{-\alpha}, \tag{2}$$

where a and $\alpha$ are constants.

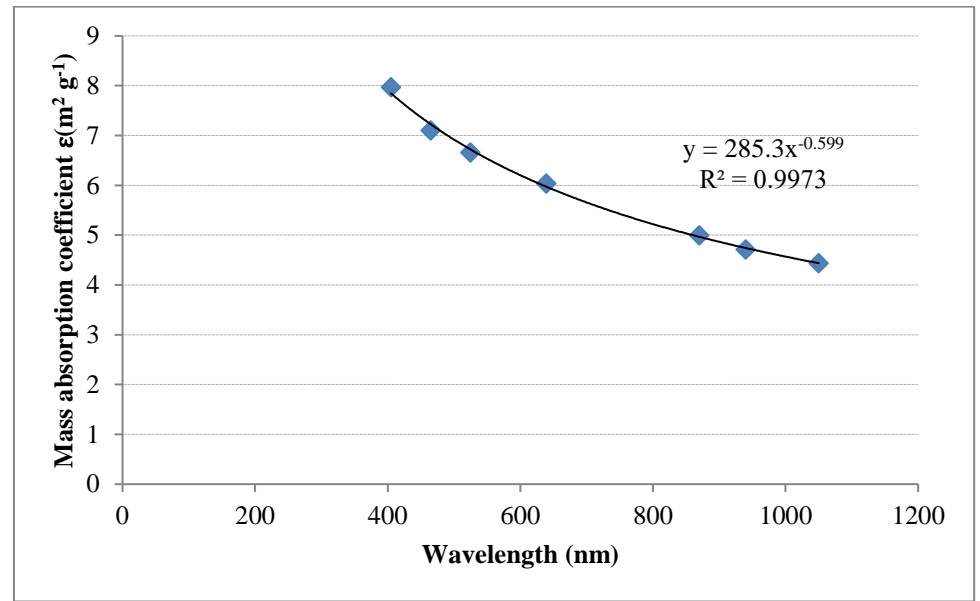

**Figure 2 The mass absorption coefficient (ε) versus wavelength (λ) obtained for sampling from the Krakow site.**

Figure 2 shows the mass absorption coefficient values for each wavelength. The fitted coefficient were: a = 285.3 and $\alpha$ = 0.599. "This value of $\alpha$ lower than unity implies that the BC particles we generally measure have core diameters in the range 150 nm to 200 nm" (David D. Cohen, 2020). The correlation was very strong with power dependence and coefficient of least squares $R^2 > 0.99$. The shape of the fit curve is as expected and mass absorption coefficient values fall within the range of 4-11 $m^2 g^{-1}$ which is consistent with the recommendations (Atanacio et al., n.d.).



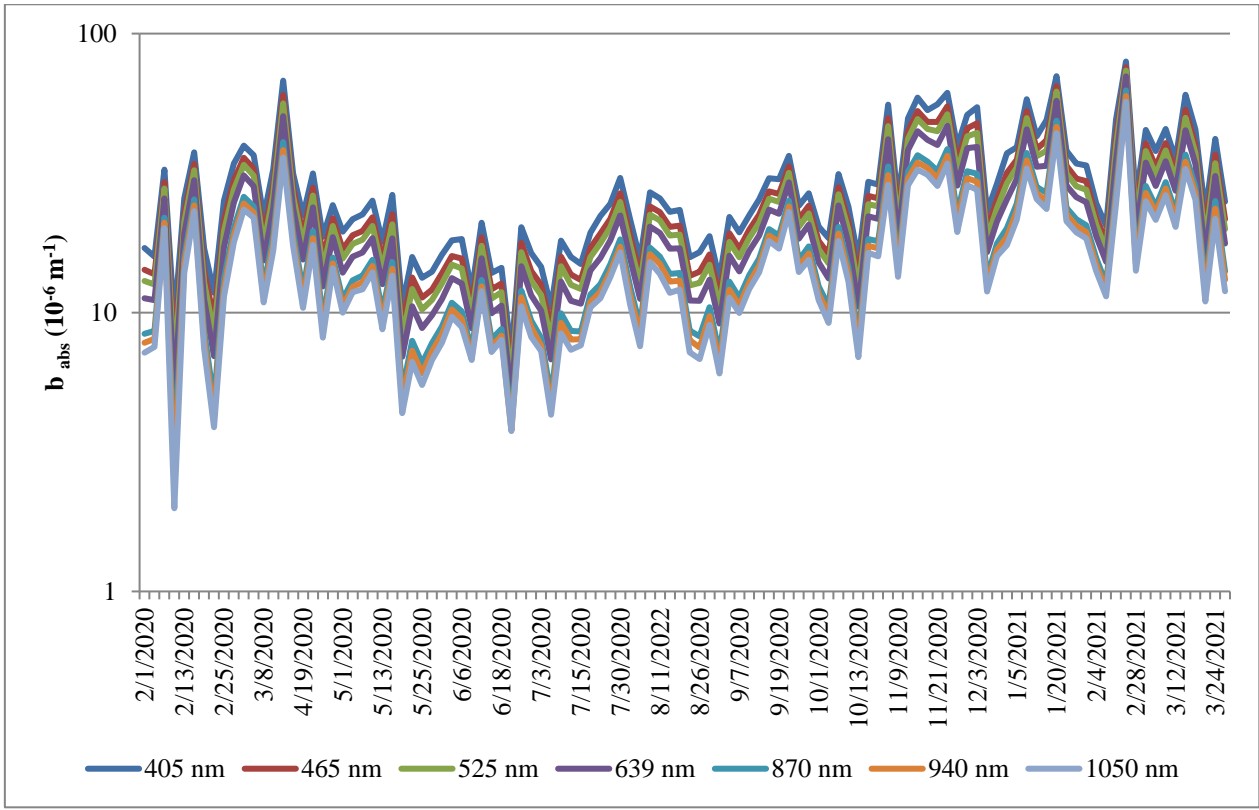

**Figure 3 The light absorption coefficients (b$_{abs}$) at seven wavelength (nm) for sampling from Krakow during February 2020 – March 2021 in 10$^{-6}$ m$^{-1}$.**

In Figure 3, the black carbon light (b$_{abs}$) fluctuated during the whole sampling time. The big differences, between February, March, November, December and June-July periods, suggest that during the cold season there was increased emission of BC for instance from combustion in a residential area, which was expected. During May, June and July, (b$_{abs}$) have smaller and more stable values. The shape of the curve and values for absorption coefficients (b$_{abs}$) can be compared to the research from Greece 2017 (Diapouli et al., 2017).





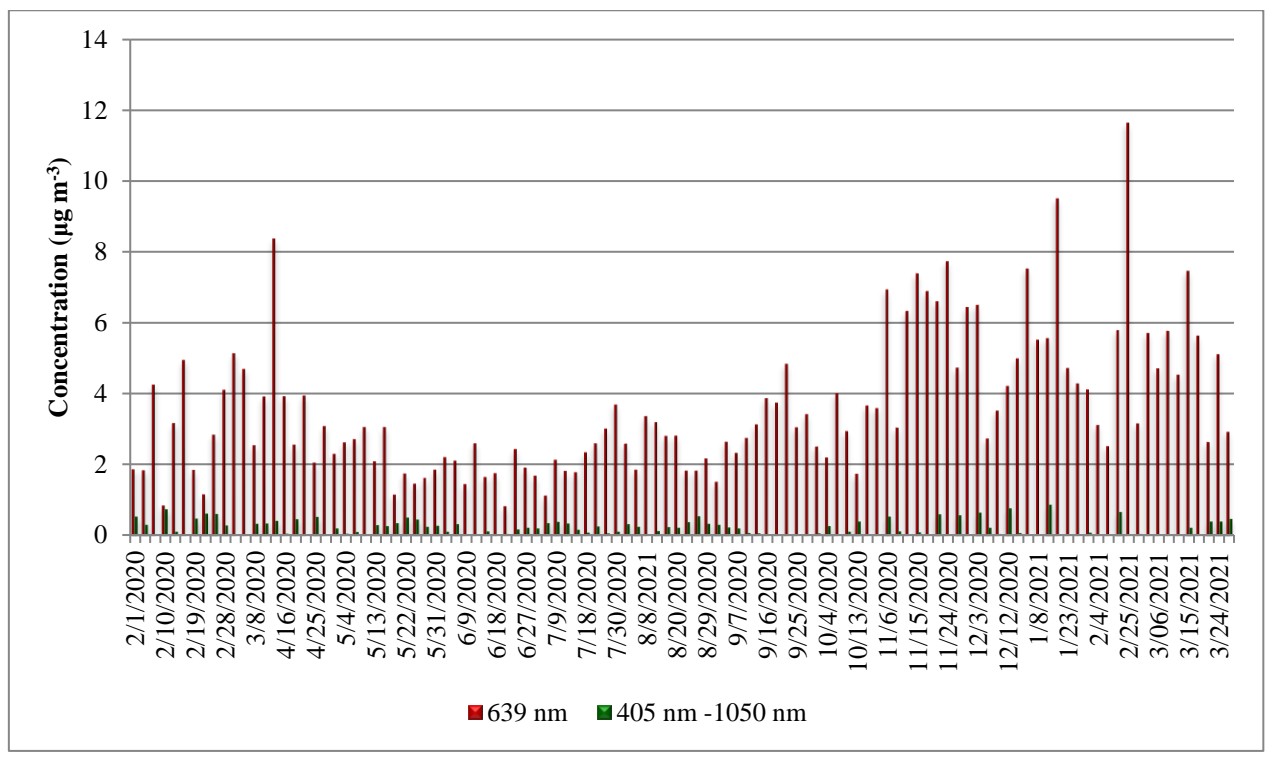


**Figure 4 Comparison daily concentrations between eBC for 639 nm and BC for (405nm – 1050nm) in μg m⁻³ for sampling from Krakow during February 2020 – March 2021.**

The red curve, in Figure 4, is the equivalent black estimate from MABI using 639 nm data. These data include, among other,
smoke components, diesel, fossil fuels, etc. The values obtained by subtracting the BC (1050 nm) data from BC (450 nm) data,
the green curve, represent mainly BC from biomass burning. It was expected that there are no significant biomass burning
event and increased pollution came from fossil fuels and transport. However, the values of BC from biomass burning were
similar to values obtained by Christian et al. in Calaca where mean values were
0.67 μg m⁻³ for winter 2019(Duc et al., 2020; Tuso et al., 2020). It can be seen in Figure 4 and Figure 5, that the values of BC
increased during March(2020, 2021), November, December (2020) and February(2021), but at May, June, July was lower and
this was the expected effect of emission from combustion. Especially, in the cold season, also there were days when the values
for eBC were particularly high. It can be explained as following, during the cold period, there is the increase of utilization the
fossil fuels among other to a heating household of coal also it can see increased traffic. Moreover, the windless weather and
low temperature are responsible for the lack of air movement and the accumulation of pollution in one place. Which causes an
increase in the concentration of $PM_{2.5}$ and eBC in the city. Interestingly, the low values of concentration BC for February 2020,
had confirmed in low concentration $PM_{2.5}$, what can suggest better weather condition compared to February 2021.
The exposed filter from 25 February 2021 (25-02-2021 in Figure 4 and Figure 8) was the blackest which is visually evident
when is compared with all exposed filters and this is a confirmation of the highest values eBC for the peak from the day.





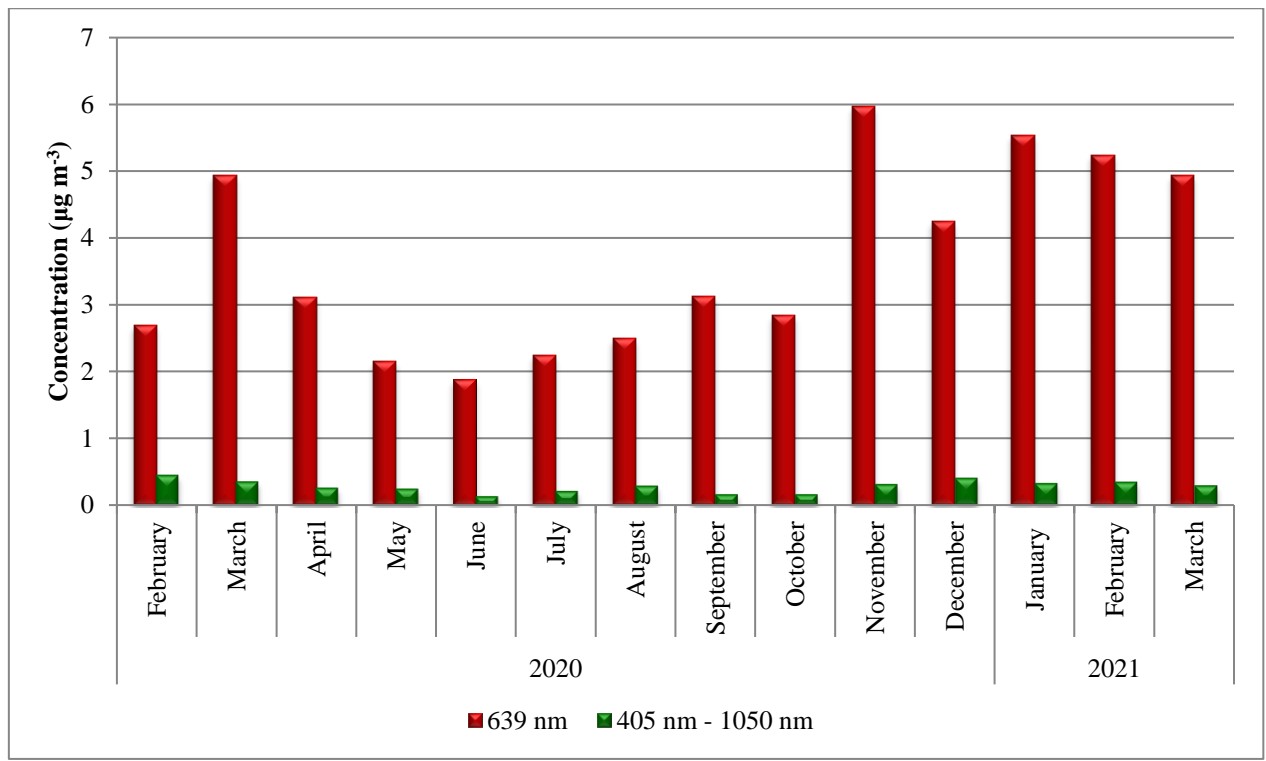


**Figure 5 Mean monthly concentrations of eBC for 639 nm and BC for (405nm – 1050nm) in μg m⁻³ for sampling from Krakow during 2020 – 2021.**

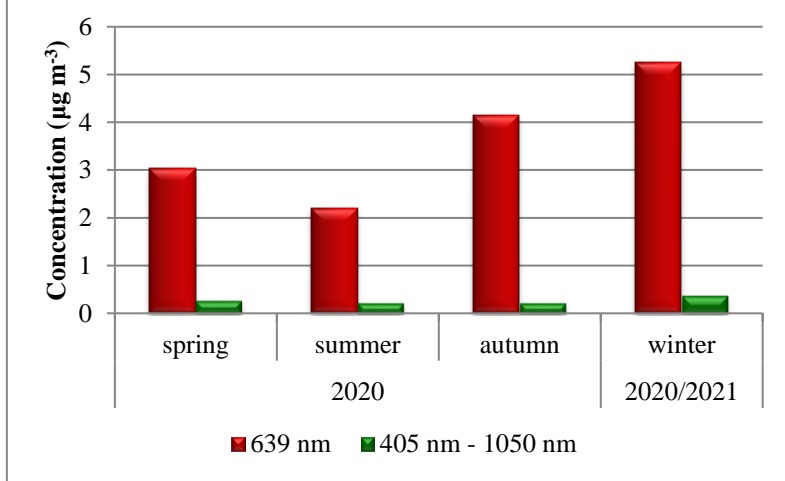


**Figure 6 Seasonal concentrations of eBC for 639 nm and BC for (405 nm – 1050nm) in μg m⁻³.**

The Figure 6 presents seasonal concentration, where seasons consist of: spring - March, April, May (2020); summer - June, July, August (2020); autumn - September, October, November(2020); winter - December(2020), January, February(2021). The mean values of eBC concentration are the highest for winter, and the lowest for summer, which was the expected outcome.





The values concentrations of BC for (405 nm – 1050 nm) can be considered constant for all seasons, which can be in connection
with biomass burning like wood etc in a household to heating water, in fireplace etc. The low value of BC for biomass and ban
of burning coal and wood in Krakow can suggested that this pollution could come not only from Krakow, but also from
neighbourhood towns which used biomass to heat the houses, heat the water etc.

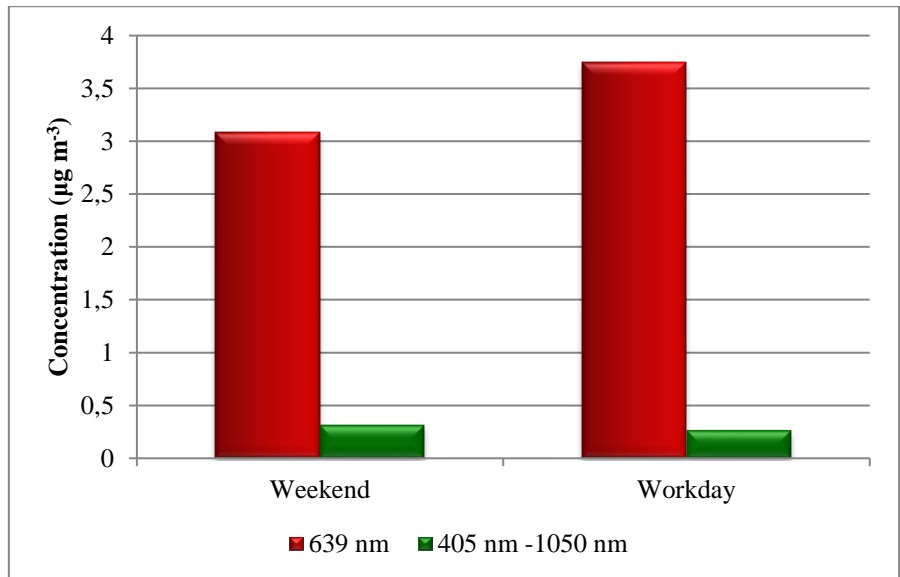


**Figure 7 Mean concentrations of eBC for 639 nm and BC for (405nm – 1050nm) for weekend and workdays**
Based on Figure 7, it can be seen that the concentration of eBC was higher for workdays than weekends. It can be in connection
with more intense traffic during workdays than at weekends. There was no significant difference between workdays and
weekends in terms of BC for (405 nm - 1050 nm).





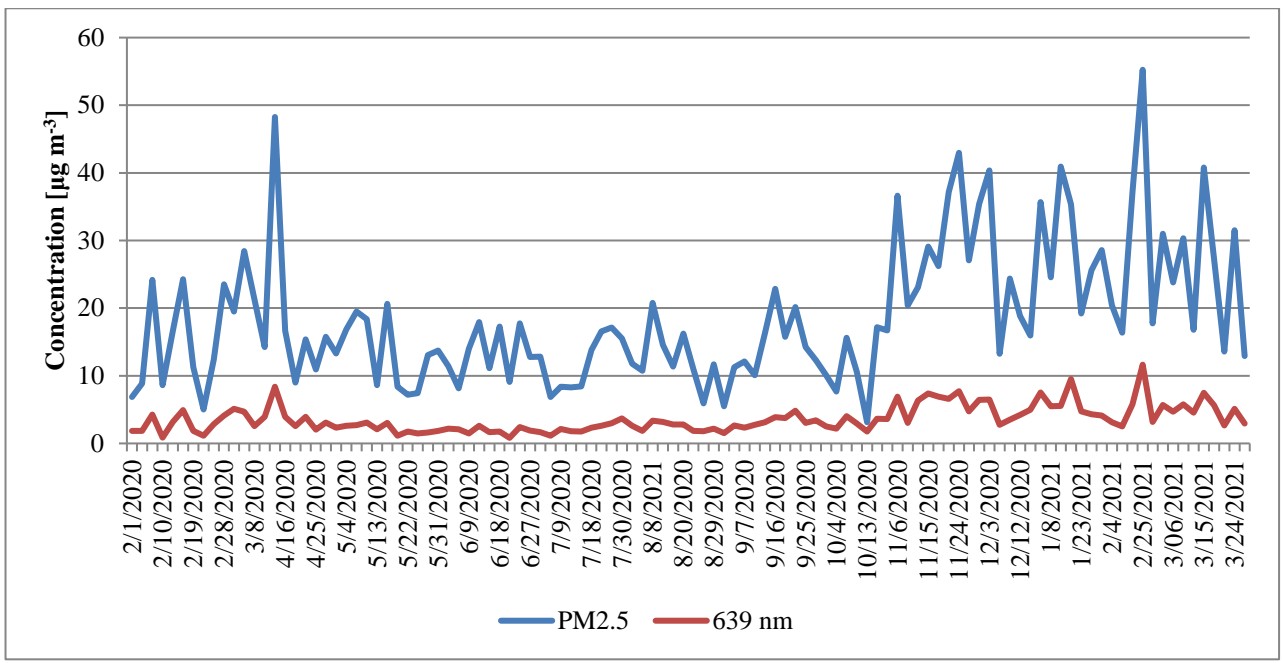


**Figure 8 Daily concentrations of PM$_{2.5}$ and eBC µg m$^{-3}$.**

Figure 8 presents the daily concentrations of PM$_{2.5}$ and eBC. As can be seen, it covers 14 months. It can be seen, that with the
concentration of the values PM$_{2.5}$ growth eBC was increasing and if values of PM$_{2.5}$ dropped then eBC (639nm) dropped too,
especially during the cold period (November, December, January, February, March). However, the concentration of PM$_{2.5}$ had
lower values in May, June, July, which was the expected effect of lower emission from combustion (also in Figure 5). In this
season, the values of eBC were more stable and seem to be kept constant despite the pick of PM$_{2.5}$ has fluctuated, which proves
that the sources of pollution were not only combustion and traffic.
Based on Table 1, it can be seen that if the PM$_{2.5}$ concentration was higher consequently the concentration of eBC was also
higher. In winter the values of PM$_{2.5}$ and eBC were picked in comparison to other seasons, and the values in summer were the
lowest what was expected.
**Table 1 Mean concentrations of PM$_{2.5}$, equivalent black carbon, black carbon related to biomass (BC$_{BB}$) in µg m$^{-3}$ with standard**
**deviation, and relative contribution of biomass (BC$_{BB}$) to eBC in % for the season, year and full sampling period.**

| PERIOD | PM$_{2.5}$ (µg m$^{-3}$) | eBC (µg m$^{-3}$) | BC$_{BB}$ (µg m$^{-3}$) | BC$_{BB}$/eBC (%) |
|---|---|---|---|---|
| **Spring** | 16.5 ± 0.2 | 3.0 ± 1.1 | 0.3 ± 0.1 | 12.3 ± 8.2 |
| **Summer** | 12.6 ± 0.1 | 2.2 ± 0.5 | 0.2 ± 0.1 | 10.8 ± 6.0 |
| **Autumn** | 19.2 ± 0.7 | 4.2 ± 1.6 | 0.22 ± 0.16 | 7.0 ± 5.1 |
| **Winter** | 27.6 ± 0.2 | 5.3 ± 1.8 | 0.4 ± 0.3 | 6.7 ± 5.5 |
| **Annual** | 18.2 ± 0.6 | 3.5 ± 1.5 | 0.25 ± 0.15 | 9.7 ± 6.5 |
| **Full period** | 18.4 ± 0.6 | 3.6 ± 1.5 | 0.3 ± 0.2 | 11.3 ± 8.4 |





In Table 1, the eBC represents the values of concentration (μg m$^{-3}$) which were obtained for λ = 639 nm; the BC$_{BB}$ - the values
were obtained for λ = 405 nm-1050 nm. The annual period of mean sampling was from March 2020 to February 2021, and the
full period – February 2020 to March 2021. It can be seen, that BC$_{BB}$ has higher values in winter than summer and autumn
which was connected with increased emission from combustion in the cold period. It can be expected, that percentage of BC$_{BB}$
could be higher in winter.
The values for eBC which were obtained in this study were similar to the result from the previous study presented among other
by Cruz et al. which was carried out during one year in 2018 and by Reche et al. in 2009 (Cruz et al., 2019; Reche et al., 2011;
Stahl et al., 2020).
Moreover, in this study, eBC was reported for 639 as is suggested by Atanacio et al.(Atanacio et al., n.d.). The analysis, which
is presented in Figure 9, shows the relationship between eBC and PM$_{2.5}$, which was confirmed by a strong correlation. Based
on this graph, it can be concluded, that about 18 % of PM$_{2.5}$ mass belongs to eBC. Similar event were also observed in other
studies where the ratio BC/PM$_{2.5}$ was about among the other 21 % and 13 % in Liverpool and Newcastle, respectively (Cohen
et al., 2000; Duc et al., 2020).

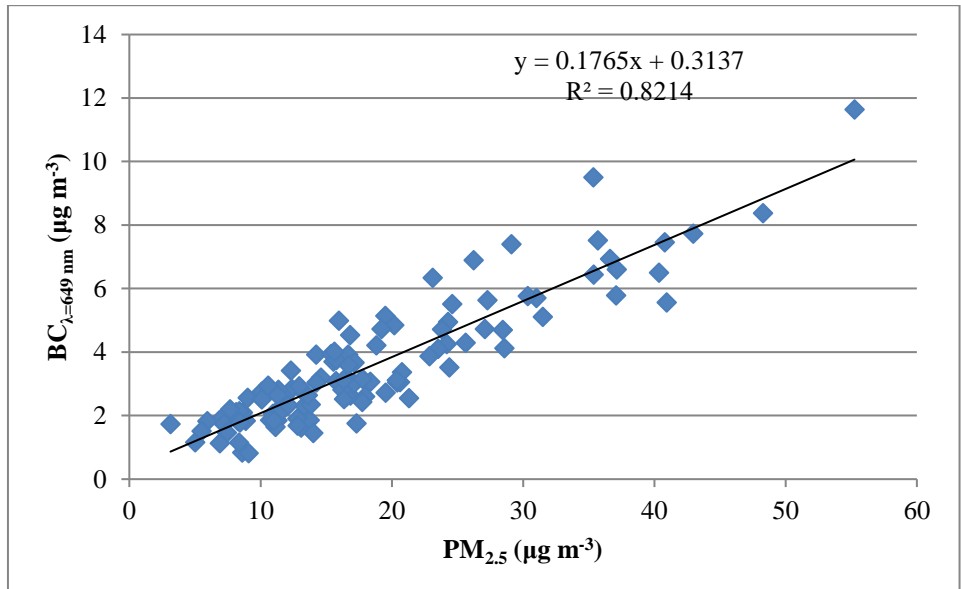


**Figure 9 Regression of concentration eBC for λ = 639 nm versus PM$_{2.5}$ for full period.**
**4 Conclusion**
To develop a strategy for lowering the concentration of black carbon in air particulate matter it is crucial to determine its
sources first. The optical absorption method allows assessing concentrations of equivalent black carbon together with the
contribution of different sources to eBC. The present study describes the methodology for the determination of eBC
concentrations and assesses the contribution of sources like fossil fuel/traffic and biomass burning to PM$_{2.5}$ mass. What is



more, we present the result with an extensive analysis of the annual period. Our work proves that fossil fuel/traffic contribution to PM$_{2.5}$ mass is higher in winter than in warm seasons. The contribution of fossil fuel/traffic is higher than biomass burning during the whole sampling period. Moreover, the concentrations of eBC and biomass burning were more stable during May, June, July than in the cold period.

**Author contribution**

AR and LS planned the experiment. AR collected samples and performed measurements and calculation. AR and LS made interpretation data, write manuscript. Both authors contributed to the paper discussion and revision.

**Competing interests**

"The authors declare no conflict of interest".

**Acknowledgements**

The International Atomic Energy Agency, project number RER/7/012 partially financed this work together with the subsidy of the Ministry of Science and Higher Education, grant number 16.16.220.842.
Research project supported/partly supported by program "Excellence initiative – research university" for the University of Science and Technology.





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
