# Peer review of "Measurement report: Determination of Black Carbon concentration in PM2.5 fraction by Multi-wavelength absorption black carbon instrument (MABI)"

_Atmospheric Chemistry and Physics, 2021_

## Referee Comment (RC1)

Review of "Measurement report: Determination of Black Carbon concentration in PM$_{2.5}$ fraction by Multi-wavelength absorption black carbon instrument (MABI)"

This study reports the black carbon concentration in PM2.5 using a MABI in Krakow, Poland from Feb 2020 to March 2021. Black carbon measurement is important given its role played in the climate as well as its negative health effects. Many methodologies have been developed to quantify black carbon including the multi-wavelength light absorption techniques (e.g., Aethalometers). This study deploys MABI using a similar light absorption principle as Aethalometers. However, I don't find any improvement in MABI in quantifying BC concentrations compared to the existing methodologies. Overall, I find this paper lacks novelties, and the results are not new although the data are new. Also, as a non-native English speaker myself, I find the English in this manuscript needs to be improved greatly to avoid confusion. For such reasons, I cannot recommend its publication in its current form and should be rejected.

Abstract. It is not clear what are the novelties of this study. You need to show new results even for a measurement report paper.

Line 13. "fossil fuel/traffic", do you mean fossil fuel or traffic, What about coal. Also, how do you separate traffic from biomass burning BC? It is not clear here.

Line 14, Do you mean the averaged fraction. Rephrase to something like "eBC, on average, accounts for 18% of PM2.5" given that "belongs to" is not scientific.

Line 17. "more" compared to what? What does "stable" mean quantitatively?

In the introduction, replace the term "scientists" with the specific studies you want to cite. As in its current form, it reads like a science book chapter or a documentary.

Line 22. I would suggest being careful with the use of the "numerous aerosol PM component". In general, the major components are organics, BC, sulfate, nitrate, etc.

Line 39, change to Bond and Bergstrom (2016)

Line 42, BC terms are also associated with measurement techniques.

Line 49, Explain the term "fraction PM2.5" or rephase. Do you mean "the PM2.5 fraction". If there are a lot of publications about BC, you should include the references following this statement.

Line 59. Your sampling period was after the ban. It is not clear what is the influence of the legislative measure regarding the eBC concentration levels and/or BC sources. How is the sampling measurement compared to previous BC measurement quantitatively?

Line 61. I am not convinced at this stage. Rephrase.

Line 62, replace the website with a proper citation.

Line 62-63, remove these sentences. The significance of this study should be presented by proper comparison with previous studies in the discussion section not in the introduction section. Replace "this year" with the specific sampling period.

Line 75, "24 hours"; what time do you start and end for a filter sampling period?

Line 78, what portion of the filter was measured for mass? For what mass, PM2.5?

Line 79, The filters were usually stored -20 °C before sampling. How the temperature of 20% and humidity of ~50% were controlled and monitored? Were there any control experiments conducted e.g., blank filter to ensure the filter sampling was quality assured with no artificial artifacts?

Line 82. I am not familiar with the instrument MABI, maybe a schematic of this instrument will provide a better idea of the sampling mechanism of this instrument. Also, can you explain the difference between the MABI and the 7-wavelength Aethalometer (e.g., AE33)?

Line 103. Can you explain why 639 nm was used for quantification? Why not 870 nm or 940 nm, where the influence from the organic fraction is less in terms of light absorption.

Figure 1. Was this figure described anywhere? What is the implication of this figure?

Line 117, specify the constants

Line 128. babs represents light absorption coefficient, not black carbon light.

Line 130. Provide a quantitative description for the "big difference", and "smaller and more stable".

Line 131. How can it be compared with the previous studies? Do you mean similar or very different?

Line 137. It is not clear how the subtraction of the BC (1050 nm) from BC (450 nm) represents BC from biomass burning.

Line 139. Why was this expected? I expect biomass burning (Atmos. Chem. Phys., 21, 14893–14906, 2021).

144-146. This is speculation. Do you have data to support your explanation?

Line 148, what is the concentration of PM2.5 in Feb 2020. What are the concentrations of PM2.5 in other months?

Line 149. Do you have any photos taken to prove your point?

Line 159, BC for biomass burning is unexpectedly low. Is this consistent with the previous study?

Line 161, What are the back trajectories like?

Figure 5-7. What are the uncertainties for the averaged values?

Technical:
Line 50: change "basing" to "based".
Line 53. Describe the equation using "where" after the equation was shown.
Line 56, add "respectively"
Line 86: rephrase "What is more, it is used the opaque glass in "MABI units to scatter the scattered light back through the filter to the detector"(David D. Cohen, 2020)."
Line 147, Which should follow ","
Line 121-122, rephrase, don't recommend using "" when you are writing a journal paper.